# Evaluating Post-Market Adverse Events of The New Hepatitis C Therapies Using FEARS Data

**DOI:** 10.3390/healthcare10081400

**Published:** 2022-07-27

**Authors:** Majed A. Algarni

**Affiliations:** Department of Clinical Pharmacy, College of Pharmacy, Taif University, P.O. Box 11099, Taif 21944, Saudi Arabia; m.alqarni@tu.edu.sa

**Keywords:** direct-acting antivirals, Harvoni, Mavyret, hepatitis C, FEARS data, pharmacovigilance

## Abstract

Background: Little is known regarding the safety of direct-acting antivirals (DAA), even though they are widely used. This study aims to evaluate the adverse events of DAA using post-market data. Methods: FDA Adverse Events Reporting System (FAERS) data from January 2019 through December 2019 were analyzed. FERAS reports in which the suspected drug contained the DAA drugs were extracted and included in the analysis. Univariable and bivariable analyses were performed in this study. Results: Most of the reported side effects were non-serious (62%). The number of times the drug was reported as ineffective was significantly higher while using Harvoni vs. Mavyret (32.14% vs. 1.05%) (*p*-value < 0.0001). On the other hand, hospitalization was significantly more reported while using Mavyret compared to Harvoni (52.02% vs. 22.45%) (*p*-value < 0.0001). Liver cancer was significantly more reported while using Harvoni vs. Mavyret (7.65% vs. 1.20%) (*p*-value < 0.0001). No significant difference in death cases was reported while using both drugs. Conclusion: Depending on the FDA Adverse Events Reporting System (FAERS) database, most of the non-serious adverse effects were headache and fatigue. There was no significant difference in cases of death reported while using both drugs. Liver cancer was more reported while using Harvoni. Hospitalization was more reported while using Mavyret.

## 1. Introduction

Hepatitis C is a liver disease caused by the hepatitis C virus (HCV). Both acute and chronic hepatitis are caused by HCV. The disease severity ranges from a mild sickness that lasts for a few weeks to a serious lifelong sickness, with approximately 399,000 individuals dying from this infection, mainly due to cirrhosis and hepatocellular carcinoma [1]. The World Health Organization (WHO) estimates that around 71 million people worldwide have chronic hepatitis C infection [1]. Direct-acting antivirals (DAAs) have changed the HCV treatment landscape, almost eliminating the older, poorly tolerated interferon treatments and offering a cure for most patients [2]. The old HCV drugs, such as peginterferon (PEG-INF) and Ribavirin (RBV), are poorly tolerated and contraindicated in a high percentage of HCV patients. The cost of the PEG-INF/RBV regimen is high, and the duration of treatment is too long (24 weeks or longer), leading to low adherence rates. The (PEG-INF)/RBV regimen is neither as selective nor as specific as the DAA [3]. The INF-Containing regimens are not cost-effective, have severe side effects, are effective mainly in the HCV1a genotype, and have several drug–drug interactions [4,5]. On the other hand, INF-Free Regimens such as NS3/4a, NS5b, and NS5a inhibitors have much better tolerability and are highly effective in all HCV genotypes [3,4,5,6]. Additionally, unlike INF-containing regimens, DAAs are taken orally and require a shorter duration of treatment (8 to 12 weeks) [7]. Effectively all patients infected with HCV are suitable for DAA therapy, including patients who are intolerant of or ineligible for interferon therapy or ribavirin. The treatment of HCV is currently approached similarly to that of HIV, with regimens consisting of combinations of drugs that target different phases of the HCV life cycle [8,9]. In recent years, several DAAs have been approved by the FDA in various combinations to interject HCV replication at different sites to prevent growth. The cure of HCV is defined by the sustained virologic response (SVR), which indicates that the virus is undetectable 12–24 weeks following the treatment. The SVR rates range from 40% to 55% in those completing treatment with dual therapy of PEG-INF/RBV, compared to 90–95% in patients treated with DAAs [10]. Still, clinicians should be aware that baseline resistance-associated substitutions (RASs) may fail treatment response to DAAs, in particular, baseline NS5A resistance in DAA-naïve HCV patients [11]. DAAs have multiple targets in the hepatitis c virus replication life cycle. NS3/4A protease inhibitors include Grazoprevir, Paritaprevir, Voxileprevir, and Glecaprevir. These drugs work by blocking the enzyme needed for the virus to grow and develop. These drugs may also weaken the virus in other ways [8,9]. Other classes of DAAs include NS5A replication complex inhibitors and NS5B polymerase inhibitors. These drugs block enzymes that are important for the virus to replicate. Most DAAs are available only in combination products, such as Epclusa (sofosbuvir and velpatasvir), Harvoni (ledipasvir and sofosbuvir), Zepatier (elbasvir and grazoprevir), Mavyret (glecaprevir and pibrentasvir), Vosevi (sofosbuvir, velpatasvir, and voxilaprevir) [12,13]. Most clinical guidelines recommend that pangenotypic DAA should be indicated to treat chronic HCV-infected individuals aged 18 years and older. The WHO and the American Association for the Study of Liver Diseases and the Infectious Diseases Society of America (AASLD/IDSA) recommendations for teens 12–17 years old or weighing at least 35 kg with chronic HCV-infected are as follows. Genotype 1, 4, 5, and 6: Sofosbuvir/ledipasvir for 12 weeks (without cirrhosis) or 24 weeks (with cirrhosis); Genotype 2: Sofosbuvir/ribavirin for 12 weeks (treatment naïve or experienced, without or with cirrhosis); Genotype 3: Sofosbuvir/ribavirin for 24 weeks (treatment naïve or experienced, without or with cirrhosis); and Genotypes 4, 5, or 6: Sofosbuvir/ledipasvir for 12 weeks (treatment naïve or experienced, without or with cirrhosis). In chronic HCV-infected children younger than 12 years, the WHO recommends deferring DDA treatment until they are aged 12 while physicians look forward to the approval and accessibility of use of DAAs for children younger than 12 years of age. However, clinical trial results of DDAs in children aged 6–12 years are starting to appear. Treatment with IFN plus ribavirin may be considered for those children with genotype 2 or 3 infection and severe liver disease [14]. The safety data of DAA have been almost entirely based on pre-market data. During the clinical trials, mild adverse reactions were reported in patients receiving DAA. To date, post-market safety data are scarce, and little is known regarding the side effects of DAA since it has been widely used in the population.

FDA Adverse Event Reporting System (FAERS) is a database that includes adverse incident reports, such as reports of medicine errors, complaints regarding products’ quality, and adverse events [15]. The database is designed to support the FDA’s post-marketing safety surveillance program for all approved medications [15]. The benefits of the FAERS public dashboard include facilitating data inquiries and producing information and charts that are easy to understand. On a regular basis, the FDA receives reports from healthcare workers, such as doctors, clinical pharmacist, and nutritionist. Additionally, consumers such as sick people, family members, or others can report to FEARS. The FEARS reporting system is a very useful tool to monitor safety concerns in marketed products. All the reports in FEARS must be checked by a professional clinical reviewer in a drug evaluation research center and at the Center for Biologics Evaluation and Research (CBER) [15].

The aim of this study is to evaluate the adverse events of DAA using post-market data by describing the serious and non-serious adverse event reports in which the suspected drug was ledipasvir-sofosbuvir (Harvoni), or glecaprevir-pibrentasvir (Mavyret). An additional aim was to assess and compare the serious adverse event reports in which the suspected drug was Harvoni with reports in which the suspected drug was Mavyret.

## 2. Materials and Methods

### 2.1. Study Design

This was a retrospective analysis of the publicly available safety data in the FDA Adverse Event Reporting System (FAERS). FAERS data from 1 January 2019 through 31 December 2019 were used in this study. Data from 2019 have been chosen because after that year, COVID-19 struck the world, and many countries enforced quarantine and other preventive measures which may have affected the way that people reported side effects [16]. Reports in which the suspected drug contained the drugs of interest were extracted and included in the study. The DAA drugs of interest in this study included Harvoni and Mavyret. A text search of generic and brand names for these drugs was performed. Reports with more than one suspected drug were excluded.

### 2.2. Study Variables

Each FAERS report contains data regarding the suspected drug, the reaction to the drug, the outcome, the type of reporter, the patient sex, the patient age, and the country where the event occurred, among others. After data extraction and data cleaning, the suspected drug was found to be either Harvoni and Mavyret. For the purpose of this study, the “reaction of the drug” variable was recoded into four categories: 1—“Drug Ineffective” when the drug had no treatment effect, 2—“Liver Cancer”, 3—“Renal failure”, and 4—“Others” for all other reported reactions. The “Outcome” variable was collapsed into three categories: 1—Hospitalized. 2—Died. 3—Other.

### 2.3. Statistical Analysis

Univariate and bivariate analysis was performed in this study. Statistical descriptive analysis of the characteristics of adverse event reports was conducted. Mean and standard deviation (SD) were produced to describe numerical variables. Frequencies and percentages were used to describe categorical variables. The percentages of serious and non-serious adverse events were calculated for each drug of interest. The proportions of adverse events that are classified as serious were compared between the two drugs of interest using Pearson’s chi-square test. A *p*-value of less than 0.05 was considered statistically significant. All statistical analyses were conducted using SAS^®^ University Edition.

## 3. Results

A total of 4899 reports were included in this study. The number of reports in which the suspected drug contained Mayvret was 3645 (74.4%) while the number of reports in which the suspected drug contained Harvoni was reported as 1254 (25.6%) (Table 1). The reports were higher in males (51.07%), while female side effect reports represented (44.32%); the rest of the patients’ reports were not specified (4.61%). The majority of reports were reported from the United States (95.28%), and only (4.72%) were reported from other countries. Most of the adverse event reports were reported by healthcare professionals (61%), and around (23%) were reported by consumers. The majority of the reported side effects were non-serious (3848 reports) (78.6%), 864 reports of them were for Harvoni, and 2984 reports were for Mavyret. Serious side effects composed 1051 (21.4%) reports, 390 of which were for Harvoni, and 661 reports for Mavyret. Among Harvoni reports, the top three frequently reported non-serious events were fatigue (83 reports) (9.55%), headache (76 reports) (8.75%), and nausea (22 reports) (2.53%). On the other hand, the most frequently reported non-serious adverse event among Mavyret reports were: fatigue (339 reports) (11.38%), headache (297 reports) (9.97%), and pruritis (97 reports) (3.26%). Table 2 shows the most frequently reported non-serious adverse events. There were 390 serious adverse events reports in which the suspected drug was Harvoni, representing 8% of all reports. Meanwhile, for Mavyret reports, there were 661 serious reports (13.5%). Figure 1 shows the percentages of serious and non-serious reports for both drugs. The bivariable analysis showed that “drug ineffective” was reported as being significantly higher while using Harvoni vs. Mavyret (32.3% vs. 1%) (*p*-value < 0.0001). On the other hand, hospitalizations were significantly more reported while using Mavyret compared to Harvoni (52.4% vs. 22.5%) (*p*-value < 0.0001). There was no significant difference in reporting renal failure: Harvoni (3%) vs. Mavyret (3.4%) (*p*-value = 0.7337). Liver cancer was significantly more reported while using Harvoni vs. Mavyret (7.6% vs. 1.2%) (*p*-value < 0.0001). No significant difference in cases of death were reported while using either drug (11.7% and 9%, respectively) (*p*-value = 0.1515). The results of the bivariable analysis between the suspected drug and the reported outcomes are shown in Table 3.

## 4. Discussion

Pharmacovigilance studies that are based on post-market safety data are important in the medical field. They are especially critical for newly approved drugs, such as DAAs, which were approved in recent years. Such studies can help us better understand these medications and protect patients from potential side effects that could not be identified in clinical trials. Safety data from pre-market clinical trials are not representative of the whole population for several reasons. First, clinical trials usually enroll a small number of participants who are not as diverse nor as representative as the target population. Second, clinical trials typically exclude special groups, such as those with pre-existing conditions. In addition, the duration of clinical trials is too short for some of the adverse events to occur. For these reasons, the FDA has always emphasized the significance and benefits of post-market drug surveillance. Because DAAs have been approved recently and have since been widely used among hepatitis c patients, this study sought to identify and describe adverse events that were not necessarily highlighted during clinical trials. In this study, headache and fatigue were the most non-serious adverse effects reported frequently among Harvoni and Mavyret users. In both previous clinical trials, it was stated that headache and fatigue were among the most common non-serious adverse events [17,18]. Pruritis was one of the top three non-serious adverse effects of Mavyret by 97 reports (3.26%), which agrees with a previous clinical trial that found that one of the common adverse events was pruritis by 8% [19]. Additionally, according to the package insert of Mavyret, it was found that pruritis forms the most common adverse effects by 17% in adults with severe renal impairment including subjects on dialysis [20]. In this study, “drug ineffective” reports among Harvoni and Mavyret reports were evaluated. Previous data indicated that there was resistance to Harvoni. Previous research found that non-adherence was the strongest risk issue for treatment failure in individuals taking sofosbuvir/ledipasvir (Harvoni) [21]. Most reasons cited for non-adherence were failing to take medication as prescribed and hospitalization [21]. Non-adherence may result in drug resistance, potentially reducing the response to ensuant therapy. A study that was conducted on HCV patients who took DAA compared with the no-HCV patients’ group found that DAA patients were more likely to have liver cirrhosis at Baseline, but after the adjustment, the risk of liver cancer compared to the non-treated group was significantly reduced [22]. Another study found that the treatment with DAAs may increase the risk of hepatocellular carcinoma, but this remains unproven [23]. This study has many limitations. First, there is an uncertainty that reported events (adverse events or medication errors) are caused by the suspected drug. Another limitation is that some reports do not have enough details to properly evaluate the adverse events. Because not every adverse event or medication error is received by the FDA, a causal relationship between a product and an adverse event cannot be established using FAERS. In addition, a duplication of the same report exists, since some reports are being submitted by both manufacturers and consumers.

## 5. Conclusions

We still do not know enough regarding the side effects of ledipasvir-sofosbuvir (Harvoni), and glecaprevir-pibrentasvir (Mavyret), even though they are widely used in the population. According to the FEARS database, most of the non-serious adverse events for both drugs were headache and fatigue. There was no significant difference in cases of death reported from using either drug. However, liver cancer was more reported while using Harvoni, while hospitalization was more reported while using Mavyret.

## Figures and Tables

**Figure 1 healthcare-10-01400-f001:**
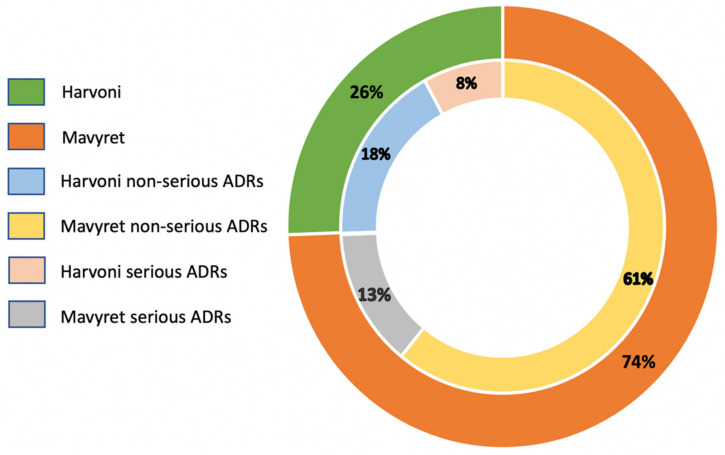
Percentages of Harvoni and Mavyret reports.

**Table 1 healthcare-10-01400-t001:** Characteristics of the study adverse event reports.

Characteristic	Total N = 4899
**Age, mean (± SD)**	56.50 (± 12.81)
**Gender, n (%)** **Male** **Female** **Not specified**	2502 (51%) 2171 (44.3%) 266 (4.6%)
**Suspected Product, n (%)** **Harvoni** **Mavyret**	1254 (25.6%) 3645 (74.4%)
**Country where event occurred, n (%)** **United States** **Other countries**	4668 (95.28%) 231 (4.72%)
**Reporter type, n (%)** **Healthcare Professional** **Consumer** **Not Specified** **Other**	2989 (61%) 1134 (23.15%) 448 (9.14%) 328 (6.7%)
**Seriousness, n (%)** **Non-serious ADR** **Harvoni** **Mavyret** **Serious ADR** **Harvoni** **Mavyret**	3848 (78.6%) 864 2984 1051 (21.4%) 390 661

**Table 2 healthcare-10-01400-t002:** Most frequently reported non-serious adverse events.

Non-Serious ADR Reports in which the Suspected Drug Was Harvoni n = 864	n (%)
Fatigue	83 (9.6%)
Headache	76 (8.7%)
Nausea	22 (2.5%)
**Non-Serious ADR Reports in which the Suspected Drug Was Mavyret** **n = 2984**	**n (%)**
Fatigue	339 (11.3%)
Headache	297 (9.9%)
pruritis	97 (3.2%)

**Table 3 healthcare-10-01400-t003:** Bivariable analysis of serious adverse events and outcomes.

Reported Outcome	Suspected Drug		*p*
	Harvoni n = 390	Mavyret n = 661	
**Drug ineffective**	126 (32.3%)	7 (1%)	<0.0001
**Hospitalization**	88 (22.5%)	347 (52.4%)	<0.0001
**Renal failure**	12 (3%)	23 (3.4%)	0.7337
**Liver cancer**	30 (7.6%)	8 (1.2%)	<0.0001
**Died**	46 (11.7%)	60 (9%)	0.1515

## Data Availability

Not applicable.

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
