# Peer review of "Evaluating Post-Market Adverse Events of The New Hepatitis C Therapies Using FEARS Data"

_healthcare, 2022, doi:10.3390/healthcare10081400_

Round 1

Reviewer 1 Report

The authors reported the adverse reactions of two medicines for treating hepatitis, Harvoni and Mavyret using the adverse reports from FERAS. Although the authors identified the differences between the two medicines, what’s value of this work to achieve better treatment?  Another concern is how the percentages (such as those listed in Table 2) were calculated. Those numbers were not calculated using the case number divided by the total reports of either drug.

Minor issues:

(1)   “The INF-Free Regimens like NS3/4a, NS5b, or NS5a inhibitor have much better tolerability and highly effective in all HCV genotypes”, in this sentence, change inhibitor to inhibitors

(2)   “.. may fail treatment response to DAA, mainly baseline NS5A resistance in DAA-naïve HCV patients [11].”, the sentence seems lack of a verb.

(3)    “Benefits of the FAERS public dashboard include facilitating data inquiries and producing easy information and charts. People who work in healthcare for example, doctors, nurses, pharmacist, and even consumers can send a report to FERAS. In regular basis, the FDA receives reports from healthcare workers, such as doctors, clinical pharmacist, nutritionist. Also, consumers such as sick people, family members or others can report to FEARS.”. The above sentences need to be concise.

Author Response

Dear reviewer, 
I would like first to thank you for your constructive comments and suggestions.

You asked about the value of this work to achieve better treatment. The aim of this study is not to determine the better treatment, but to evaluate the safety of the drugs based on post-market data, which may help determining the best therapeutic plan for the patient.

You also asked about the percentages in Table 2. The percentages were calculated by dividing the number of non-serious adverse event of concern (e.g., fatigue) divided by the total number of reports of non-serious adverse events of the drug (864 reports for Harvoni, and 2984 reports for Mavyret). Those detailed numbers of reports were not added in the previous manuscript, so I added them now in the text, and in Table 1 and Table 2.

Finally, all your minor language suggestions were addressed in the revised manuscript.

I highlighted the new sentences in Yellow for easier tracking.

Thank you again for your helpful comments and suggestions.

Best regards,

Reviewer 2 Report

Thank you for giving us the opportunity to review this manuscript.

It is a good work overall, but here are some specific comments

Introduction and method

There is so much details about the uses of the drugs in certain population which in my opinion are not necessary. More Focus on Harvoni and Mavyret would be more appropriate. I would’ve expected to see more details about post-marketing adverse events systems, especially FAERS should be moved to the introduction

 Please cite this sentence

2019 data has been chosen because after that year 89 COVID-19 struck the world and many countries enforced quarantine and other preventa- 90 tive measures which may affected the way that people reported side effects

Discussion is well written

Conclusion

Please add the names of the drugs in the opening sentence

Author Response

Dear reviewer, 
I would like first to thank you for your constructive comments and suggestions.

You asked for a citation for a sentence in the manuscript, a citation is added for this sentence in the revised manuscript.

You also asked for the names of the drugs to be added in the revised manuscript, and they were added.

thank you again for your constructive suggestions,
Best regards,

Round 2

Reviewer 1 Report

The authors have adequately addressed my concerns.